# Fibromyalgia Syndrome Pain in Men and Women: A Scoping Review

**DOI:** 10.3390/healthcare11020223

**Published:** 2023-01-11

**Authors:** Ilga Ruschak, Pilar Montesó-Curto, Lluís Rosselló, Carina Aguilar Martín, Laura Sánchez-Montesó, Loren Toussaint

**Affiliations:** 1Internal Medicine Unit, Sant Pau i Santa Tecla Hospital, 43003 Tarragona, Spain; 2Faculty and Department of Nursing, Rovira i Virgili University, 43003 Tarragona, Spain; 3Primary Care in Institut Català de la Salut (ICS), 43500 Tortosa, Spain; 4Department of Medicine and Surgery, Rovira i Virgili University, 43201 Reus, Spain; 5Rheumatology Unit, Fibromyalgia and Chronic Fatigue Syndrome Unit Coordinator, Santa Maria Hospital, 25198 Lleida, Spain; 6Research Support Unit, University Institute for Primary Care Research (IDIAP Jordi Gol), 43500 Tortosa, Spain; 7Evaluation Unit, Primary Health Care Terres de l’Ebre Department, Institut Català de la Salut, 43500 Tortosa, Spain; 8Physical Medicine and Rehabilitation, Università di Roma Tor Vergata, Policlinco Tor Vergata, 00133 Rome, Italy; 9Department of Psychology, Luther College, Decorah, IA 52101, USA

**Keywords:** fibromyalgia, pain, assessment, experience, review

## Abstract

Fibromyalgia syndrome (FMS) is a chronic musculoskeletal disorder of unknown etiology that affects up to 5.0% of the world population. It has a high female predominance, between 80 and 96%. Due to the low number of diagnosed men, research work has focused mainly on women. The extensive body of literature on sex differences in pain in the general population suggests that men and women differ in their responses to pain, with greater sensitivity to pain and a higher risk of clinical pain commonly observed among women. This review aims to: (1) determine how pain is assessed or what types of questionnaires are used, (2) examine whether there are differences in pain characteristics between men and women with FMS and (3) describe how pain is conceptualized or manifested in patients at a qualitative level. In this study, the scoping review method of articles published in the last 5 years (2016–2022) was used. Ten articles were included. The most used questionnaires and scales to assess pain were the PVAS (Pain Visual Analogue Scale) and the FIQ (Fibromyalgia Impact Questionnaire). On the other hand, five categories were obtained: (1) qualities of pain, (2) uncertainty and chaos, (3) pain as an aggravating factor, (4) adaptation to the new reality and (5) the communication of pain. It has been observed that both subjective perception and widespread pain are higher in women. Men, on the other hand, have a worse impact of the pathology, more painful experiences and more catastrophic thoughts about pain. An updated knowledge of pain in FMS and whether it differs according to sex would be beneficial for clinicians to make an earlier diagnosis and treatment and, in turn, benefit patients suffering from this chronic disease.

## 1. Introduction

Fibromyalgia syndrome (FMS) is a chronic musculoskeletal disorder of unknown etiology that affects up to 5.0% of the world’s population [1,2]. The incidence is greater in Europe (2.64%) than in America (2.41%) or parts of Asia (1.62%) [3]. The percentages fluctuate from country to country because the ways of determining them are diverse, as are the age groups included and the differentiations in sociocultural standards. Consequently, for example, the prevalence in Spain is about 2.4%, while in the USA it is 2% [4]. This pathology greatly alters individual health-related quality of life. The vast bulk of affected people end up suffering from diverse kinds of disability, isolation, stigmatization, lack of validity of their diagnosis and concern about their long-term prognostication [5].

Fibromyalgia syndrome has a high female predominance, accounting for 80–96% [4,6]. Still, a systematic review of FMS in men and women worldwide described that the predominance of the condition is similar for both sexes, i.e., approximately 3.98% in women and 2.40% in men [7]. Due to the low number of diagnosed men, research has mainly focused on women, ignoring the study of this syndrome in men. The difference between men and women in the prevalence and diagnosis of FMS appears to be related to the social stigma related with it being a mostly female illness and to the social and cultural characteristics of Western countries, where men are less likely to go to a specialist for chronic pain symptoms, which limits the formulation of a correct diagnosis [8,9].

The extensive body of the literature on sex disparities in pain in the general population strongly suggests that men and women differ in their responses to pain, with greater pain sensitivity and higher risk of clinical pain commonly observed among women [10,11]. Thus, the idea of sex differences in FMS symptoms gains clarity. If women and men with FMS present with a different impact and intensity of symptoms, it would be advisable to diagnose and treat on an individualized basis.

Fibromyalgia syndrome presents with a wide variety of signs and symptoms, making it difficult to diagnose. So far, there are no biomarkers of the disease and clinical evaluation and patient descriptions are used. Continuous efforts have been made to improve the diagnostic accuracy of FMS [12]. The 2016 American College of Rheumatology criteria are the most accurate and used in clinical practice. According to the Wolfe et al. definition, FMS is a condition that involves widespread pain in at least four of five regions (left upper region, right upper region, left lower region, right lower region, axial region) and the symptoms must have been present for at least 3 months. Also, the widespread pain index (WPI) must be equal to or greater than seven and the symptom severity scale (SSS) score must be equal to or greater than five. Alternatively, the integrated pain management (IPM) must be between four and six and the SSS score must be equal to or greater than nine. Furthermore, the diagnosis is valid independently of other diagnoses; i.e., FMS does not exclude the presence of other clinically important diseases [13].

As for treatment, it is not curative, and its aim is to reduce symptoms in order to provide greater functionality to the person [14]. As it is a very complex condition, all authors conclude that the treatment of FMS should be holistic, comprehensive and with a multidisciplinary approach [15].

Pain is the central symptom of FMS, which coexists with many other symptoms such as fatigue, insomnia, cognitive dysfunction and mental health disorders [16]. Fibromyalgia syndrome pain is defined as chronic, meaning that it persists or recurs for more than three months [17]. It mainly affects the musculoskeletal system and is present throughout the body, from the head to the feet [18]. Usually, initially the pain is localized, but over time it affects many muscle groups. It is characterized by being persistent with variable intensity, while it can often be described as a burning sensation or stabbing pain. There is oversensitivity to normally painful stimuli, such as pressure or heat (hyperalgesia) and painful sensation to normally non-painful stimuli, such as touch (allodynia) [19].

The fact that FMS presents chronic pain without any obvious peripheral tissue damage has given rise in recent years to the new concept of nondisciplastic pain, also known as nocioperception, which comes from the Latin nocere: pain that activates peripheral nociceptors without clear evidence of actual or threatening tissue damage.

The type, location and severity of pain depend on several modulating factors, the most important of which are physical exercise, comorbidities such as obesity and temperature variations [20].

The present scoping review has focused the study of pain in men and women in FMS. Considering that it is a disabling symptom that is present daily in people with FMS, we believe that it is of vital importance to find out its characteristics known to date and thus be able to provide a more updated view to clinicians (especially in Primary Care) to provide earlier diagnosis and treatment and, in turn, benefit patients suffering from this chronic disease.

Our aims focused on: (1) determining how pain is assessed or what types of questionnaires are used, (2) examining whether there are differences in pain characteristics between men and women with FMS and (3) describing how pain is conceptualized or manifested in the participants at a qualitative level.

## 2. Materials and Methods

This scoping review was undertaken in line with the Preferred Reporting Items for Systematic Reviews (PRISMA) guidelines for scoping reviews [21]. The Preferred Reporting Items for Scoping Reviews (PRISMA-ScR) Checklist can be found in Appendix A.

The revision protocol registration number is 10.37766/inplasy2022.12.0105, available at https://inplasy.com/inplasy-2022-12-0105/.

### 2.1. Inclusion Criteria

The identified studies were subjected to inclusion and exclusion criteria. To be included, studies had to be published from January 2016 to July 2022, available in full text, written in English or Spanish and use both quantitative (observational studies) and qualitative methodology. Clinical trials, case studies, opinion articles, interventions, or systematic reviews (with or without meta-analysis) were excluded to ensure higher quality evidence.

### 2.2. Search Strategy

The bibliographic search was carried out during the months of February and July 2022. The electronic databases used for the search were PubMed, SCOPUS, CINAHL, Web of Science and Google Scholar. In each of these, an exhaustive search was performed using a combination of Boolean logic and truncations for the following keywords: “pain”, “fibromyalgia”, “men”, “women”, “conceptualization”, “manifestation”, “score” and “assessment”.

The following search string was used for SCOPUS: (fibromyalgia) AND (pain) AND (men OR women) AND (score OR manifestation OR concept* OR assessment). For the PubMed database, we used ((((“Fibromyalgia”(Mesh)) AND “Pain”(Mesh)) AND “Men”(Mesh)) OR “Women”(Mesh)) OR (“Pain Measurement/classification”(Mesh) OR “Pain Measurement/instrumentation”(Mesh) OR “Pain Measurement/nursing”(Mesh) OR “Pain Measurement/psychology”(Mesh)) OR (score) OR (concept*) OR (manifestation) OR (assessment). For the CINAHL, Web of Science and Google Scholar Boolean we used the terms fibromyalgia AND pain AND (men OR women) AND (score OR manifestation OR concept* OR assessment. The complete search strategy is illustrated in Figure 1.

### 2.3. Selection of the Studies

The studies obtained were imported and processed using the bibliographic reference management software Mendeley Desktop^®^ version 1.19.4 (London, UK).

The selection process consisted of two levels of screening of the articles obtained: (1) a review of the title and abstract and (2) a review of the full text.

The articles retrieved by the database search were evaluated by the PhD supervisor, who made the evaluation by reading the title and abstract provided by the PhD student. The previous authors also read the full texts of all publications that could not be excluded at the title/abstract level. They reviewed the abstracts/titles of the articles and agreed on which met the inclusion/exclusion criteria for full-text review. Disagreements about study selection and data extraction were resolved by consensus by the majority of the authors of the present review or by using an external author [22].

### 2.4. Data Extraction

Data were extracted from full-text articles that met the objectives and inclusion criteria. Definitive data were obtained using a data extraction form with the following information: author(s), year of publication, country, study design, study objective(s), participants (sex and age range). In the case of quantitative studies, the types of questionnaires used were also considered. In the qualitative articles, for the review of the experiences of men and women with FMS, the authors proceeded to analyze the discourses related to the conceptualization or manifestation of pain by means of categories and subcategories.

### 2.5. Process Followed to Determine the Categories and Subcategories of Pain in the Qualitative Studies

We progressed with two phases to obtain and show the presence in the verbal explanations and experiences of people with FMS of the categories and possible subcategories in each study: (1) statements, responses, individual or group aspects related to pain were identified; (2) after having recognized the categories, a second in-depth analysis was carried out that allowed us to get subcategories and to be able to catalogue them [23].

## 3. Results

Our search strategy resulted in a total of 10 final references after the selection process, as can be seen in Figure 1.

**Figure 1 healthcare-11-00223-f001:**
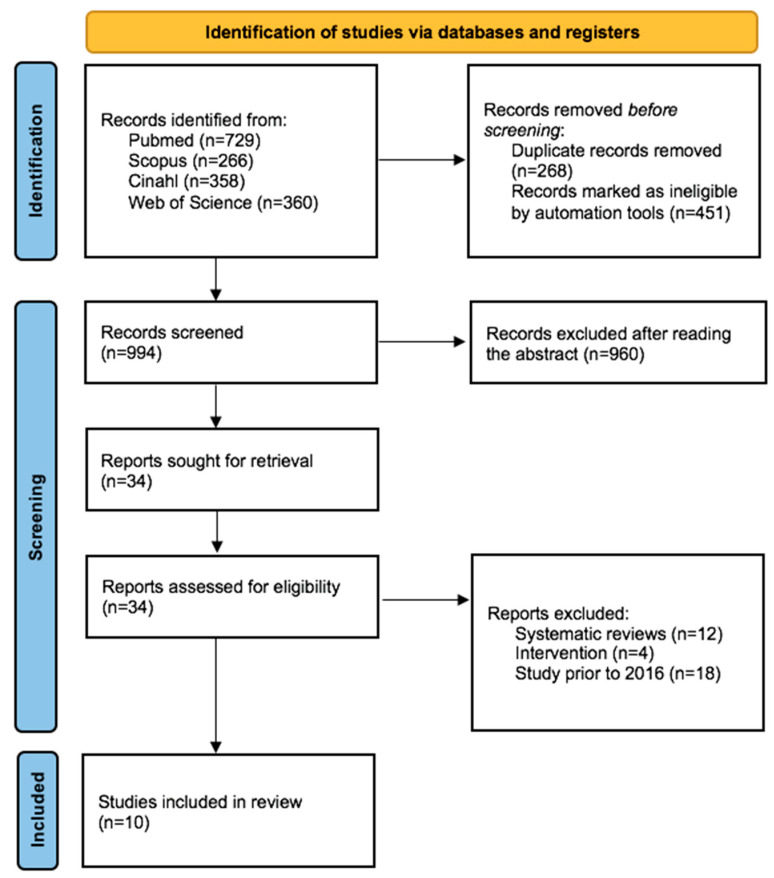
Flow diagram.

All selected studies with their main results are shown in Table 1.

Of the 10 studies that met the inclusion criteria, 7 were quantitative [1,24,25,26,27,28,29], 2 qualitative [16,31] and 1 mixed (qualitative methodology together with quantitative methodology) [30]. Sample sizes ranged from 5 to 4342 participants. The sum of all participants with FMS from the studies was 5222 (2262 males and 2960 females); of these, the female sex predominated with 56.7%. In all studies, the age range was 18 years or older; older subjects had a mean age of 56.6 ± 12.6 years.

Two quantitative studies worked only with women [27,28] and two studies featured only male participants; one qualitative [16] and one mixed [30]. The other six studies worked with mixed-sex participants [1,24,25,26,29,31].

The predominant countries were Spain with three studies [24,28,30] and the United States with two [25,30]. The study by Kueny et al. was performed in two different countries, Spain and the United States, with the aim of being able to observe pain in different cultural contexts [30]. The patients came from the United States [25,30], Spain [24,27,29,31], Finland [16], France [27], Thailand [26], China [1] and Italy [29]. The Spanish study by Sendra and Farré contained patients from all over the world, as they worked with the Instagram social network [31].

The pain assessment of the selected quantitative studies included a wide variety of questionnaires and scales (Table 2) such as the Fibromyalgia Impact Questionnaire (FIQ), total number of Tender Points (TP), Pain Catastrophizing Scale (PCS), Widespread Pain Index (WPI), Symptom Severity Scale (SSS), Polysymptomatic Distress (PSD) and Pain Visual Analogue Scale (PVAS). The most used were the PVAS (out of 10 or out of 100) and the FIQ.

On the other hand, qualitative studies used tools such as focus groups, interviews and narrative and life history to describe how pain manifests itself in men and women with FMS.

The PVAS and WPI were higher in female participants [1,25,29]. In Wolfe’s study, males were older (64.9 ± 12.0 years) than females (59.7 ± 13.5 years) [25]. In contrast, in the Jiao study men were significantly younger *p* = 0.027 (43.6 years) than women (50.1 years) [1]. The FIQ and PCS were somewhat higher in men than in women in the Segura-Jiménez study, [24] but without statistical significance. There were no differences for the rest of the scales.

The review of the qualitative literature allowed us to detect the presence of categories related to pain. A total of five categories were obtained: (1) qualities of pain, (2) uncertainty and chaos, (3) pain as an aggravating factor, (4) adaptation to the new reality and (5) communication of pain.

In the first category—pain qualities—we integrated the results associated to the pain characteristics reported by patients in the chosen studies. The second category—uncertainty and chaos—describes how patients cope with chronic pain. The third category—pain as an aggravating factor—reflects the different causes of chronic pain in patients. In the fourth category—adaptation to the new reality—we show how patients make vital changes to continue living with pain. Finally, in the last category—pain communication—we show the importance of expressing emotions and sharing the experience of pain as a benefit for the person.

### 3.1. Pain Qualities

In the study by Sallinen and Mengshoel men affected by FMS vividly described their pain with fluctuating intensity from day to day [16]. On good days the pain was almost non-existent, but on bad days it could become agonizing. On the other hand, the male participants in Kueny’s study also described the pain as excruciating, a “pain [through] the soul”. Yet, their worst pain was reported as shooting and location-specific rather than being described as widespread or generalized as we are accustomed to reading about in fibromyalgia patients [30].

### 3.2. Uncertainty and Chaos

The patients in the Sendra and Farré study manifest continuous uncertainty when suffering from an incurable disease that manifests with chronic pain. Everyday actions become major obstacles [31]. Since this pathology is sometimes difficult to diagnose, most patients used narratives of chaos to talk about their chronic pain. They feel they are losing control of their lives and, at the same time, this loss of control ends up affecting their identity. These patients mostly reflect that they do not perceive a positive evolution of their health, although a small percentage are confident about their future despite all their problems. In the context of this disease, the existence of this duality is manifested in chronic diseases as they tend to oscillate between periods of exacerbation, in which symptoms such as pain worsen, and periods of quiescence, in which disability is less disturbing.

If we focus only on male patients, this uncertainty also manifests itself; Kueny described that men were surprised by the sporadic nature of their pain, a constant pain that is ever present in their lives and makes it difficult for them to make plans [30].

### 3.3. Pain as an Aggravating Factor

Selected studies show that patients are not only threatened by fibromyalgia pain but also by the pain as an aggravator of other symptoms [16,30]. Patients describe problems falling asleep, which in turn causes tiredness and daytime fatigue [30]. Pain and their lack of energy limit them in all levels of their social relationships [31]. They describe having changes in their personality, feeling more irritable with others and even being disappointed with themselves for not being able to fulfil their roles as they would like or as they did before [30].

According to Sendra and Farré, patients with chronic pain end up isolated and stigmatized, as presenting “invisible pain” causes them emotional distress by having to repeatedly demonstrate their disability to others, both people close to them and healthcare professionals [31]. The study by Kueny also reflects how difficult the invisible nature of the pain they experience is for their patients and they also acknowledged having to be their own advocates in the face of others who did not believe them [30].

### 3.4. Adaptations to the New Reality

People suffering from chronic pain tend to seek a balance between health and illness, between ability and disability [16]. According to the selected qualitative studies, patients tend to look for which are the aggravating factors of their pain in order to decrease or eliminate them; for example, with physical exertion such as walking, patients explained that the more they moved, the more pain they experienced; consequently, they had to manage their body movements well during the day [30]. Patients expressed having to make major changes in their lives in order to “live with their pain”: changes in work, role, personal life, family, routines, etc. [31].

### 3.5. Communication of Pain

The previous categories reflect how chronic pain causes drastic changes in the lives of people who suffer from it, and although pain is presented as the main enemy of these people, studies reflect that it is vitally important to talk about it. Sendra and Farré suggest that sharing the experience is very beneficial, either self-reported, i.e., verbally, or with the help of assessment tools. It should be kept in mind that suffering chronic pain is a recurrent experience with clearly emotional components that have to be drained in an individualized way for the biopsychosocial well-being of the person [31].

## 4. Discussion

This review provides an updated synthesis of fibromyalgia pain from different approaches to better understand it.

As for the way in which pain has been assessed, the results show that the PVAS (which assesses the subjective perception of global pain) and the FIQ (which assesses the impact of illness) were the most used tools. Despite this, we believe that the consideration of the PVAS in terms of its structure and the patient’s behavior on the scale casts doubt on its validity. It is linear and is prone to biases; for these reasons, we consider that its use should be limited. On the other hand, we are confident in the use of the FIQ, as it assesses the status, progress and prognosis of patients and is an instrument that is extensively used in the healthcare setting [32].

We contrasted whether there were differences in pain characteristics according to sex. We detected that the subjective perception of pain was higher in women, as was the generalized pain index (WPI). In contrast, FMS impact was higher in men, as were painful experiences and PCS pain thoughts. The worst Pain Visual Analogue Scale scores are obtained by women both in studies analyzing only women [27,28] and in studies studying men and women [1,25,29]. The worst FIQ scores are those obtained by Kueny in male patients with FMS [30]. In Ubeda-D’Ocasar’s study only with women, the highest PVAS values were present in the supraspinatus muscle, the trochanteric prominence and the upper outer quadrant of the buttocks, respectively [28].

Wolfe’s study with men and women exposed low PVAS values in both sexes and no significant differences [25]. The Chinese study by Jiao also did not detect significant differences between the PVAS of their men and women [1]. In contrast, the mixed study by Iannuccelli showed high PVAS values in women with statistical significance [29], although it should be noted that in all three studies the representation of men with respect to women was very low.

The FIQ values in the mixed study by Segura-Jiménez [24] were elevated, but there were no significant differences among men and women with FMS. In contrast, in the Italian study by Iannucceli, women had higher FIQ values than men [29]. Úbeda-D’Ocasar’s study [28] only with women *(n* = 30 W) also presented high values with the FIQ, although, the values were higher in the male study *(n* = 17 M) by Kueny and colleagues. The FIQ was also compared by country (USA or Spain) to detect whether there were cultural contrasts in different health policies. Spanish men had higher values, but the differences only approached statistical significance. Demographic factors do not directly influence pain perception but represent valuable individual difference factors [30]. Although several examples of epidemiological evidence have shown that chronic pain conditions are more predominant among women than among men, [35,36] in our review we only detected this in Jiao’s study, which revealed significant differences in pain according to sex. Women with FMS had worse values in the WPI questionnaire (generalized pain) and men with FMS had worse values in the SSS (severity of symptoms) [1].

Finally, through the use of qualitative methodology this review wanted to describe how pain is conceptualized or manifested in affected individuals. Pain, according to male participants in the studies by Sallinen and Mengshoel [16] and Kueny, has qualities of being fluctuating. Also, agonizing and unbearable at the worst times. Unlike other studies, the pain of the men in Kueny et al. is of a localized type, not generalized and has a stabbing characteristic, such as by an arrow or sword [30]. According to Ruschak’s study, the pain of men with FMS was also described as “like an arrow or heart attack” and of a fluctuating type; i.e., it was present in different sections of the body, not generalized [37]. The chronicity and random nature of fibromyalgia pain causes much uncertainty and chaos in the lives of these patients [30,31]. Fatigue and insomnia also entail major problems in their lives, which are closely linked to pain, which is their direct aggravating factor [30], a phenomenon also shared in the study of Ruschak et al. [37]. These in turn also limit them at all levels of their social relationships [30,31] because the invisible nature of pain means that patients have to be their own advocates in front of others, because they do not believe them. These challenging situations full of negative attitudes have been previously described in other studies on FMS [9,37,38,39]. According to Ruschak and colleagues, the lack of understanding shown by some clinicians, as well as their family and friends, has had a very bad impact on patients’ health, principally psychological health [37]. Sallinen suggests that there is also a consequence for their identity, especially their masculinity, as it has had to be renegotiated and reconstructed [16]. All these changes in their lives are difficult to face, but necessary. Acceptance of the new reality helps people to move on, mainly with the help of others. This help begins with communicating their discomfort and finding a receptive listener so that they can talk about their pain [40].

## 5. Conclusions

The results of this review provide updated information on FMS pain in both sexes. To date, we can see that pain remains a very complex, internal and private sensory experience and more so in men because FMS is still mostly conceptualized as a women’s disease.

It has been observed in a few studies that both subjective perception and the generalized pain index are higher for women, but a worse impact, more painful and more severe experiences, and also more catastrophic thoughts about pain in men should be considered. In any case, the results have little statistical significance, and we consider that it is necessary to increase the sample of men in the studies so that these particularities can be studied in greater depth.

To improve pain care in these patients, we believe that there is a need for multidisciplinary management including educational interventions aimed at health care personnel on the diverse concepts of pain (subjective perception, impact of pain, widespread pain, localized pain, severity of symptoms, catastrophic thoughts about pain), to thus help to improve the understanding of individual and gender disparities in pain.

The results of this review have been made possible by the increasing inclusion of men with FMS and the awareness that the male experience and perspective is just as important as the female. Even so, we encourage further expansion of the male sample in future studies, because with the current results the significative differences in male and female pain did not reach statistical significance in all the studies, probably due to the small sample of men.

## 6. Limitations

Some limitations must be mentioned. First, reducing the review to the last 5 years allowed us to identify that there are fewer studies than we thought related to the subject. Secondly, in the mixed studies, the proportion of men to women is unbalanced and the low male representation is detrimental to them and limits their perspective. Finally, there are biases in some regions and we detected that there are countries in which FMS is not studied as much, for example Asian countries. This is probably due to the type of healthcare access they have or their cultural beliefs.

In short, FMS remains an area that needs more awareness and investigation by researchers.

## Figures and Tables

**Table 1 healthcare-11-00223-t001:** Selected studies with their main findings.

Reference (Country) Study Type	Aim	Sample	Questionnaires and Scales	Findings
Segura-Jiménez et al., 2016 (Spain) [24] Comparative Cross-Sectional Study **Quantitative**	To examine gender differences in sensitivity, fibromyalgia impact, health-related quality of life, fatigue, sleep quality, mental health, cognitive performance, pain cognition and positive health in Spanish fibromyalgia patients and non-fibromyalgia individuals of the same age and region. To observe the optimal cut-off score of the different sensitive items for women and men.	**FM patients = 388** W = 367 M = 21 **No FM patients = 285** W = 232 M = 53	**Tender Points (0–18)**	FM-W: 16.8 ± 0.1 FM-M: 16.8 ± 0.4 ***p* = 0.877 NS** No FM-W: 3.3 ± 0.2 No FM-M: 0.8 ± 0.4 ***p* < 0.001** Women reported greater pain sensitivity
			**Fibromyalgia Impact Questionnaire (0–100)**	FM-W: 64.7 ± 0.9 FM-M: 65.5 ± 3.6 ***p* = 0.837 NS** No FM-W:20.7 ± 0.9 No FM-M:18.7 ± 1.9 ***p* = 0.339 NS**
			**Pain Catastrophizing Scale** **(0–52)**	FM-W: 25.1 ± 0.7 FM-M: 26.2 ± 2.7 ***p* = 0.712 NS** No FM-W: 11.2 ± 0.7 No FM-M: 9.9 ± 1.5 ***p* = 0.427 NS**
Wolfe et al., 2018 (EE. UU.) [25] Longitudinal Study **Quantitative**	To compare CritFM with ClinFM to investigate gender and other biases in fibromyalgia diagnosis.	**FM patients = 4342** W = 2171 M = 2171 Age = 56.6 ± 12.6 years W = 59.7 ± 13.5 years M = 64.9 ± 12.0 years	**Widespread Pain Index (0–19)**	FM-W:5.9 ± 0.7 FM-M:4.9 ± 1.3
			**Symptom Severity Scale (0–12)**	FM-W = 4.3 ± 0.7 FM-M = 3.4 ± 1.1
			**Polysymptomatic Distress (0–31)**	FM-W = 10.2 ± 1.6 FM-M = 8.2 ± 1.6
			**Pain Visual Analogue Scale (0–10)**	FM-W = 3.9 ± 0.3 FM-M = 3.4 ± 1.0
			Higher values of pain and symptom severity were detected in women relative to men. Since FMS is defined based on pain and symptom severity, women will always be more likely to be diagnosed. In short, there is a relationship between being female and being diagnosed with FMS.	
Prateepavanich et al., 2018 (Thailand) [26] Cross-Sectional Study **Quantitative**	To obtain demographic data, clinical characteristics and investigate correlations of clinical features in Thai patients with FMS.	**FM patients = 71** W = 69 M = 2 Age = 44.83 (±10.81) years	**Pain Visual Analogue Scale (0–100)**	63.39 ± 17.8
			**Fibromyalgia Impact Questionnaire (0–100)**	45.48 ± 16.83
De Roa et al., 2018 (France) [27] Comparative Cross-Sectional Study **Quantitative**	To characterize childhood experiences, perceived lack of parental affection, hypersensitivity to stimuli, life stressors, anxio-depression and ergomania.	**FM-W patients = 44** **Migraine-W patients= 34** Age = 45 ± 12 years	**Pain Visual Analogue Scale (0–10)**	Better moments: FM-W = 3.3 ± 1.9 Migraine-W= 1.8 ± 2.3 Worse moments: FM-W = 8.9 ± 1.4 Migraine-W= 8.7 ± 1.2 NS Scores
Jiao et al., 2021 (China) [1] Cross-Sectional Study **Quantitative**	To characterize the demographics, severity of fibromyalgia-related symptoms and quality of life (QoL) among Chinese fibromyalgia patients.	**FM patients = 124** FM-W = 107 FM-M = 17 Age-W = 50.1 years Age-M = 43.6 years P = 0.027 M significantly younger Mean age= 49.4 years	**Pain Visual Analogue Scale (0–100)**	FM-W: 56.2 ± 21.7 FM-M: 54.1 ± 25.5 ***p* = 0.72 NS**
			**Widespread Pain Index (0–19)**	FM-W = 11.1 ± 4.7 FM-M = 8.6 ± 3.9 ***p* = 0.038** Women higher values of pain
			**Symptom Severity Scale (0–12)**	FM-W = 7.4 ± 2.6 FM-M = 8.7 ± 1.8 ***p* = 0.06** Males higher values of symptom severity
			**Polysymptomatic Distress (0–31)**	FM-W = 18.5 ± 5.9 FM-M = 17.2 ± 4.6 ***p* = 0.40 NS** No gender differences in either group
Úbeda-D’Ocasar et al., 2021 (Spain) [28] Descriptive Exploratory Study **Quantitative**	To assess the pain pressure thresholds (PPT) and subjective pain perception (SPP) of the 18 PTs while applying standardized pressure.	***n*= 30 W** Mean age = 55.1 ± 8.7 years	**Fibromyalgia Impact Questionnaire (0–100)**	FM-W: 64.1 ± 14.4 Nine locations were examined bilaterally: TP1 forehead; TP2 intertransverse space of C5-C7; TP3 midpoint of the trapezius muscle; TP4 supraspinatus muscle; TP5 second costochondral junction; TP6 2 cm distal to the lateral epicondyle; TP7 upper outer quadrant buttocks; TP8 trochanteric prominence; TP9 in the medial fat of the knee.
			**Pain Visual Analogue Scale (0–100)**	The most painful points located in: TP7: 69.6 ± 19.4 TP8: 68.0 ± 21.5 TP4: 65.1 ± 21.1 The lowest points located in: TP5: 1.28 ± 0.42 TP1: 1.52 ± 0.34 TP8: 1.61 ± 0.59 ***p* > 0.05 NS**
Iannuccelli et al., 2022 (Italy) [29] Cross-Sectional Study **Quantitative**	To evaluate the influence of gender on clinical manifestations, with special attention to the neuropsychiatric features of FMS.	***n* = 172 W** ***n* = 29 M** Mean age = 49.13 years	**Pain Visual Analogue Scale (0–10)**	FM-W = 7.5 ± 1.64 FM-M = 6.52 ± 2.06 ***p* = 0.0130**
			**Fibromyalgia Impact Questionnaire (0–100)**	FM-W = 68.07 ± 16.06 FM-M = 55.17 ± 18.26 ***p* = 0.0005**
			**Widespread Pain Index (0–19)**	FM-W = 10.67 ± 3.91 FM-M = 10.90 ± 4.81 ***p* = NS** No gender differences in either group
			**Symptom Severity Scale (0–12)**	FM-W = 9.24 ± 1.72 FM-M = 8.724 ± 1.79 ***p* = NS** No gender differences in either group
Kueny et al., 2021 (EEUU, Spain) [30] **Mixed** **(****1.** **Quantitative****)**	To describe the pain and fatigue experiences of men with MFS from Spain and the United States.	***n* = 17 M** Spain-M = 10 USA-M = 7 Age range = 30–63 years Mean Age = 52 years	**Fibromyalgia Impact Questionnaire (0–100)**	Spain M: 81.93 ± 5.89 USA M: 67.99 ± 15.33 ***p* = 0.08** The difference only approached statistical significance.
**(2.** **Qualitative** **)**	To describe the pain and fatigue experiences of men with MFS from Spain and the United States.	***n* = 17 M** Spain-M = 10 USA-M = 7 Age range = 30–63 years Mean Age = 52 years	**Focus groups and interviews**	Common experiences (Spanish and American) include **fluctuating pain** (especially with movement), pain considered **invisible** to others and localized pain. **Pain triggers**, such as thermosensitivity. Physical exertion, such as walking. Both samples acknowledged that the more they moved, the more pain they experienced.
Sallinen and Mengshoel, 2017 (Finland) [16] **Qualitative**	To elucidate the impacts of FMS on men’s daily life and work capacity.	***n* = 5 M**	**Life story**	**Major changes** in their work, hobbies and diet to control symptoms, such as pain. Participants recognized the **importance of physical activity** and struggled to find an activity that did not aggravate aches and pains.
Sendra and Farré, 2020 (Global) [31] **Qualitative**	To identify how and why patients use online platforms for pain communication.	** *n* ** **= 350 M and W**	**Narrative**	**Sharing the painful experience** can be beneficial for patients, because chronic pain brings constant problems and disbelief. **Illness narratives** allow patients to explain this condition in new ways. However, the **lack of time in doctor–patient interactions** hinders the use of this intervention for communication by increasing the **communication gap**. With the **Internet era**, patients have sought **other venues** to express their concerns in online settings. Patients often do not disclose their disease to avoid **stigmatization and disbelief** when interacting with others.

FM: Fibromyalgia, W: Women, M: Men, NS: Not significant.

**Table 2 healthcare-11-00223-t002:** Questionnaires and scales used to measure outcomes in selected studies.

N°	Category	Questionnaires and Scales
1	Total number of Tender Points	TP (0–18): Patients were considered to have fibromyalgia if they had 11 or more positive tender points [24].
2	Fibromyalgia Impact Questionnaire	FIQ (0–100): Comprises 21 individual questions with a rating scale of 0 to 10. These questions comprise three different domains: function, overall impact and symptoms score (ranging 0–30, 0–20 and 0–50, resp.). The FIQR total score ranges from 0 to 100, with a higher score indicating greater impact [32].
3	Pain Catastrophizing Scale	PCS (0–52): Was used to assess painful experiences and thoughts or feelings about pain. It contains 13 items on a 5-point scale. For this study, the total score (ranging from 0 to 52) was used, where higher score represents a more negative appraisal of pain [24].
4	Widespread Pain Index	WPI (0–19): The widespread pain index is a summary count of the number of 19 painful regions, a self-reported list of painful regions [29].
5	Symptom Severity Scale	SSS (0–12): The symptom severity scale is the sum of the severity scores of three symptoms (fatigue, waking without rest and cognitive symptoms) (0–9) plus the sum (0–3) of the number of the following symptoms that have bothered the patient and occurred during the previous 6 months: (1) headaches (0–1), (2) lower abdominal pain or cramps (0–1) and (3) depression (0–1 [29]).
6	Polysymptomatic Distress	PSD (0–31): The polysymptomatic discomfort scale (known as FM severity), is the sum of the WPI and SSS. The PSD measures the magnitude and severity of FM symptoms [33].
7	Pain Visual Analogue Scale	PVAS (0–10) or (0–100): Assesses the subjective perception of global pain (from 0, no pain, to 10 or 100, maximum pain) [34].

TP: Tender Points, FIQ: Fibromyalgia Impact Questionnaire, PCS: Pain Catastrophizing Scale, WPI: Widespread Pain Scale, SSS: Symptom Severity Scale, PSD: Polysymptomatic Distress, PVAS: Pain Visual Analogue Scale.

## Data Availability

No new data were created or analyzed in this study. Data sharing is not applicable to this article.

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
