# Peer review of "Fibromyalgia Syndrome Pain in Men and Women: A Scoping Review"

_healthcare, 2023, doi:10.3390/healthcare11020223_

Round 1

Reviewer 1 Report

The paper is a reasonably well-analyzed meta-analysis.. but we don't learn anything new. But i'm not in principle opposed to publishing a paper like this. The paper gives a clear perspective. So, I leave it to the Editor. My opinion is neutral. 

I have one comment for the authors-- many of these subjects might (men and women) might be consuming alcohol for pain moderation. And they would not have explicitly mentioned it for stigma-reasons.  Do you think this may affect the men/women discrimination effects you observe?  

Author Response

I attach the answers in the file. Thank you.

Reviewer 2 Report

The numeration of the pages is wrong, please fix it.

ABSTRACT

- P. 1 line 37. You can also focus on the clinical application. 

INTRODUCTION

- It could be interesting to include more information about the prevalence or  socio-demographic data of the Spanish and the USA populations.

METHODS

- P 3, line 105. It could be interesting to complete the Preferred Reporting Items For Systematic Reviews and Meta-Analyses para Scoping reviews (PRISMA-ScR), for facilitating the review process of the article.

- P 3, line 104. Please, could you specify the registration number of the review protocol in the methods section?

-P 3, line 108. Please, could you clarify me why the search has been limited to 2016-2022? There are previous studies that could be interesting to include in the review.

- P 3, line 128. You have mentioned that the complete search strategy is illustrated in Figure 1, but the figure 1 doesn´t appear in any section of the article. Please, could you add it?

RESULTS

- I think is not necessary to put the references in two formats when the included articles are mentioned.

- P 4, line 159.  It could be interesting to use the new PRISMA flowchart. 

- Table 1. If the article of Segura-Jiménez et al. compared FM patients with no FM patients, it would be interesting to add the comparation between them in the findings section of the table.

- Table 1. It could be interesting to include the findings of Wolfe et al. as longitudinal study, not only the comparation between male and female patients. Also, the p values should be added.

- Table 2. Please, could you add the footnote with the different abbreviations?

DISCUSSION

- In my opinion, the firsts parts of the discussion sections need to be summarized, or at less, more comparations with previous reviews/studies or other pathologies should be added in these parts.

- Please, could you add a specific section about the possible limitations of your study?

Author Response

(The authors gave the same response as above.)

Round 2

Reviewer 2 Report

Congratulations for the article again, The topic of the article is so interesting, I´m sure that the research applications will be of interest for this approach. I am very grateful for all your answers. However, some comments need to be realized:

- P 3, line 104. Please, could you specify the registration number of the review protocol in the methods section?

 Response: We have registered the protocol with INPLASY, but it takes about 48 hours to obtain the number... Would it be possible to send the number later? Thank you and sorry for the inconvenience.

Reviewer´s Response: I appreciate that the review has been registered, but I believe that it should not be published until it has been approved for registration... I hope you understand.

Why the registration has not been realized in PROSPERO? 

Thank you for your answer.

 - P 4, line 159.  It could be interesting to use the new PRISMA flowchart. 

Response: We have used the PRISMA flowchart, if you can now see figure 1, you will be able to appreciate it.

 Reviewer´s Response: It is not the last format of flow diagram of PRISMA, here is it:

https://prisma-statement.org : “PRISMA 2020 flow diagram”

Author Response

Dear Reviewer,

Thank you very much for your help. Please find attached the file with the answers. 

Best regards,

All authors. 
